# FUNCTIONAL CRITIC MODELING FOR PROVABLY CONVERGENT OFF-POLICY ACTOR-CRITIC

## ABSTRACT

Off-policy reinforcement learning (RL) with function approximation offers an effective way to improve sample efficiency by reusing past experience. Within this setting, the actor–critic (AC) framework has achieved strong empirical success. However, both the critic and actor learning is challenging for the off-policy AC methods: first of all, in addition to the classic "deadly triad" instability of off-policy evaluation, it also suffers from a "moving target" problem, where the policy being evaluated changes continually; secondly, actor learning becomes less efficient due to the difficulty of estimating the exact off-policy policy gradient. The first challenge essentially reduces the problem to repeatedly performing off-policy evaluation for changing policies. For the second challenge, the off-policy policy gradient theorem requires a complex and often impractical algorithm to estimate an additional emphasis critic, which is typically neglected in practice, thereby reducing to the on-policy policy gradient as an approximation. In this work, we introduce a novel concept of functional critic modeling, which leads to a new AC framework that addresses both challenges for actor-critic learning under the deadly triad setting. We provide a theoretical analysis in the linear function setting, establishing the provable convergence of our framework, which, to the best of our knowledge, is the first convergent off-policy target-based AC algorithm. From a practical perspective, we further propose a carefully designed neural network architecture for the functional critic modeling and demonstrate its effectiveness through preliminary experiments on widely-used RL tasks from the DeepMind Control Benchmark.

## 1 INTRODUCTION

**Off-policy AC.** Reinforcement learning (RL) has proven to be a powerful tool for sequential decision-making. In this work, we focus on *off-policy* RL under the actor-critic (AC) framework. In contrast to on-policy methods, off-policy methods enjoy improved sample efficiency by reusing past data collected by a behavior policy (White, 2015; Sutton et al., 2011; Lin, 1992; Mnih et al., 2015; Schaul et al., 2016). The actor-critic framework (Degris et al., 2012; Maei, 2018; Silver et al., 2014; Lillicrap et al., 2015; Wang et al., 2016; Gu et al., 2017; Konda, 2002) is arguably one of the most successful frameworks for policy-based control, which is favored in applications with continuous and/or high-dimensional action spaces, such as robotics and large language models.

**Challenges.** Despite the empirical success of off-policy AC, this framework still faces several challenges—both in practice and in theory. One widely-studied difficulty lies in the critic update step, which conducts off-policy evaluation for the target policy. In contrast to the on-policy setting, applying standard evaluation techniques such as temporal-difference (TD) learning with function approximation under the off-policy setting can lead to instability—a phenomenon known as the "deadly triad" problem (Baird et al., 1995; Tsitsiklis and Van Roy, 1996; Kolter, 2011; Sutton and Barto, 2018). Although there are convergent off-policy evaluation methods under the deadly triad, such as gradient TD methods (Sutton et al., 2009; Qian and Zhang, 2025; Yu, 2017; Maei et al., 2009), emphatic TD methods (Sutton et al., 2016; Yu, 2015), and target critic-based methods (Mnih et al., 2015; Lee and He, 2019; Zhang et al., 2021; Chen et al., 2023), provable convergence within the off-policy AC framework remains largely unresolved, due to two additional challenges in critic learning and actor learning respectively. First, under the AC framework, the evaluated policy is constantly changing, as the actor is updated continuously—leading to a moving target problem for the off-

policy evaluation/critic learning. In fact, even for on-policy AC, existing convergence guarantee depends on a two-timescale framework (Konda, 2002), where the policy has to be updated much more slowly than the updates driving the critic's convergence. Second, the off-policy policy gradient (Imani et al., 2018) is more complex than its on-policy counterpart, as it involves not only value estimation but also correction terms, known as the emphatic term, to account for distribution mismatch. These corrections introduce additional sample-based estimation steps to critic learning, which is vulnerable to the same instability issues as the policy-evaluation step (Zhang et al., 2020).

**State-of-the-art.** To address the instability issue and additional challenges of off-policy actor–critic (AC), Zhang et al. (2020)—to our knowledge the only prior work with a convergence guarantee—combines two types of provably stable gradient-based evaluation methods, known as gradient TD (GTD) (Sutton et al., 2009) and Emphatic TD (ETD)(Sutton et al., 2016) within a two-timescale framework. While this ensures convergence, it substantially slows policy improvement, since most samples are consumed by critic updates rather than policy updates. Moreover, the GTD and ETD-based algorithm is complicated, requiring estimation of several additional critic-like quantities that introduce extra computational and numerical burdens as well as provable instability concerns under practical settings (Manek and Kolter, 2022). In practice, most successful off-policy RL algorithms (Fujimoto et al., 2018; Haarnoja et al., 2018; Lillicrap et al., 2015; Ball et al., 2023) use target critics to stabilize TD learning, as this approach has been empirically shown to be easier to tune than gradient TD methods. However, because of the two additional challenges noted above in off-policy AC, none of the target-based methods is provably convergent under function approximation. In particular, since off-policy policy gradients require extra sample-based estimation for the emphatic term—and the corresponding target-based estimation techniques remain unclear—target-based AC methods often fall back to on-policy gradient formulas, avoiding algorithmic complications but resulting in less efficient policy improvement and lacking the convergence guarantee of exact off-policy gradients. Moreover, to address the moving-policy issue, state-of-the-art empirical methods (Chen et al., 2021; Ball et al., 2023) typically employ a high update-to-data (UTD) ratio, which mirrors the two-timescale design but likewise reduces sample efficiency.

**Our solution.** To overcome the challenges faced by existing off-policy AC methods, we introduce a new actor-critic framework with *functional critics* that eliminates both the need for slow-rate policy updates and the need for correction terms in exact off-policy gradients. The key idea is to model the critic as a functional that maps the policy space to policy values, for given state or state-action pair. For policy evaluation, this allows the critic to generalize to new, changing policies without needing to restart the evaluation process after every actor update. For policy improvement, exact off-policy gradients can be directly computed from the learned functional critic, without correction terms and additional estimation loops. In learning functional critics, we adopt a functional TD learning scheme that leverages the existence of the Bellman equation for each changing policy.

To support our framework with theoretical insights, we analyze it in a setting under *linear functional approximation*—an adaptation of the standard linear value function approximation widely-used by existing theoretical work, to the functional setting (see §3.2). This simplified model enables us to rigorously demonstrate that our framework yields the first convergent off-policy target-based AC algorithm—without relying on a multi-timescale design between actor and target-critic updates (key updates driving the critic's convergence)—thereby validating our insights of functional critic modeling and highlighting the potential of our framework to address fundamental challenges in off-policy AC.

Our implementation of the functional critic leverages the expressive power of modern neural networks to approximate complex mappings between policies and value estimates (Vaswani et al., 2017; Radford et al., 2018; 2019; Dosovitskiy et al., 2020; Devlin et al., 2019). In §4, we present a minimal implementation of our proposed framework, incorporating only the theoretical insights without any bells and whistles. In two widely used continuous control tasks from the Deepmind Control Benchmark(Tassa et al., 2018), we report preliminary empirical results that compare favorably to those of a representative state-of-the-art off-policy AC method. This highlights the potential of our framework, particularly given that our implementation omits several widely adopted heuristics considered essential for existing off-policy AC methods.

**Summary of contributions.** This work makes three key contributions,

1. The novel concept of functional critic modeling, which leads to a new AC framework that addresses the two major challenges for actor-critic learning under the deadly triad setting.

2. The first provably convergent off-policy target-based AC algorithm under function approximation.

3. A minimal implementation of the proposed AC framework using modern neural networks that achieves encouraging preliminary results.

## 2 BACKGROUND

**Markov Decision Process.** We consider an infinite-horizon *Markov Decision Process* (MDP) with a finite state space $\mathcal{S}$ with $|\mathcal{S}|$ states, a finite action space $\mathcal{A}$ with $|\mathcal{A}|$ actions, a transition kernel $p : \mathcal{S} \times \mathcal{S} \times \mathcal{A} \rightarrow [0, 1]$, a reward function $r : \mathcal{S} \times \mathcal{A} \rightarrow \mathbb{R}$, and a discount factor $\gamma \in [0, 1)$. At each time-step $t$, after an action $a_t$ is picked, agent then proceeds to a new state $s_{t+1}$ according to $p(\cdot|s_t, a_t)$ and gets a reward $r(s_t, a_t)$. Given any policy $\pi : \mathcal{S} \times \mathcal{A} \rightarrow \Delta(\mathcal{A})$, we define

$$V_\pi(s) := \mathbb{E}[\sum_{t=0}^\infty \gamma^t r(s_t, \pi(s_t))|s_0 = s], \quad Q_\pi(s, a) := r(s, a) + \gamma \mathbb{E}_{s_1}[V_\pi(s_1)|s_0 = s, a_0 = a],$$

where the expectation above are taken with respect to all future transitions and the randomness of $\pi$. Given an initial distribution $d_0$ of states, the goal here is to find a policy $\pi$ that maximize the expected value under $\mu_0$, defined as $J(\pi; d_0) := \sum_{s \in \mathcal{S}} d_0(s)V_\pi(s)$.

**Off-Policy Reinforcement Learning.** In off-policy RL, the learner does not have the knowledge of the underlying transition kernel $p$, but can interact with the environment using some *behavior policy* $\mu$. If we denote the stationary distribution of states under $\mu$ by $d_\mu$ [1], the goal of off-policy RL is to approximate the policy $\pi^\star$ that maximize $J(\pi; d_\mu) := \sum_s d_\mu(s)V_\pi(s)$. For simplicity of the notation, we assume the behavior policy $\mu$ is fixed as in previous off-policy AC analysis (Zhang et al., 2020; Khodadadian et al., 2021). throughout the paper and denote $J(\pi; d_\mu)$ by $J(\pi)$. Throughout the paper, we parametrize the policy $\pi$ by a parameter $\theta \in \Theta \subseteq \mathbb{R}^d$ and denote the value function and $Q$-function under the policy $\pi_\theta$ by $V_\theta$ and $Q_\theta$, respectively.

**The Actor-Critic Framework.** The actor-critic (AC) framework is a widely used approach in off-policy reinforcement learning, consisting of two key components: the *actor*, which selects actions according to a parameterized policy $\pi_\theta$, and the *critic*, which evaluates this policy by estimating $V_\theta$ or $Q_\theta$. Among various algorithms within this framework, policy gradient methods, are commonly employed to update the actor in off-policy setting, as detailed below:Among various algorithms within this framework, policy gradient methods, are commonly employed to update the actor in off-policy setting, and the critic then is applied to evaluate gradients, as detailed below:

**(a) Critic update.** One core challenge in the critic's update step arises from the instability of temporal-difference (TD) learning when combined with function approximation in the off-policy setting—a phenomenon known as the *deadly triad*. More precisely, given any policy $\pi$ to be evaluated, while TD learning is simple and effective in on-policy settings, it is provably unstable in off-policy setting when function approximation are used together due to the mismatch of $d_\mu$ and $d_\pi$. This instability has motivated a line of research on alternative methods for off-policy evaluation, including gradient-based TD methods (Sutton et al., 2009; Zhang and Whiteson, 2022; Qian and Zhang, 2025), Emphatic TD methods (Sutton et al., 2016; Yu, 2015), and target-network based methods (Chen et al., 2023; Zhang et al., 2021; Fujimoto et al., 2018; Haarnoja et al., 2018; Lillicrap et al., 2015).

**(b) Actor update.** For the actor, estimating the policy gradient in an off-policy setting is further complicated by the distribution mismatch. Specifically, based on chain rule, we have

$$\nabla_\theta J(\theta) = \sum_{s \in \mathcal{S}} d_\mu(s) \sum_{a \in \mathcal{A}} \left[ \nabla_\theta \pi_\theta(a|s) Q_\theta(s, a) + \pi_\theta(a|s) \nabla_\theta Q_\theta(s, a) \right]. \quad (1)$$

The key challenge lies in calculating the second term of (1): unlike in the on-policy setting, where $d_\mu(s)$ is replaced by $d_{\pi_\theta}(s)$ so that $\sum_{s \in \mathcal{S}, a \in \mathcal{A}} d_{\pi_\theta}(s) \pi_\theta(a|s) \nabla_\theta Q_\theta(s, a) = 0$ (Sutton et al., 2000), in off-policy setting $\sum_{s,a} d_\mu(s) \pi_\theta(a|s) \nabla_\theta Q_\theta(s, a)$ is in general non-vanishing since $d_\mu \neq d_{\pi_\theta}$.

---

[1] Such stationary distribution exists under some structural assumptions on the underlying MDP.

From a practical perspective, directly ignoring the second term in (1) (i.e. using on-policy gradient approximation) has led to some empirical success (Degris et al., 2012; Silver et al., 2014; Lillicrap et al., 2015; Fujimoto et al., 2018) as well as local policy improvement guarantees (Degris et al., 2012). Theoretically, however, to ensure the convergence of off-policy AC, an unbiased estimation of the exact formula (1) is needed. Existing work relies on a policy-dependent *emphasis* term $m_\theta(s) := \mathbb{E}_{a_t \sim \pi_\theta(s_t)}[\sum_{t=0}^{+\infty} d_\mu(s_t)|s_0 = s]$, to evaluate the following exact equivalent formula (Imani et al., 2018) of (1),

$$\nabla_\theta J(\theta) = \sum_s m_\theta(s) \sum_a \nabla_\theta \pi_\theta(a|s) Q_\theta(s, a). \tag{2}$$

The policy-dependent emphasis term $m_\theta(s)$ reflects state dependent reweighting to correct for the sampling distribution. Like the value function, this emphasis term must be estimated from data, and is similarly prone to the instability issues mentioned above, which has to be carefully treated and combined with convergent off-policy evaluation techniques in order to have provably convergent estimates (Zhang et al., 2020). However, this additional estimation step introduces extra learning-rate schedules, which complicates hyperparameter tuning and prevents the method from scaling as effectively as prior empirical approaches, thereby creating a mismatch between empirically successful algorithms and those with provable convergence guarantees.

Finally, we conclude this section by highlighting an additional challenge in developing provably convergent algorithms for target-based off-policy AC, which is the central focus of Section 3.2. While Zhang et al. (2021); Chen et al. (2023) demonstrate that incorporating target network–based designs can provably resolve the off-policy $Q$-evaluation problem, their policy improvement guarantees are established only *value-based control* rather than for actor–critic. The key difficulty is that an estimator of $Q$ values alone is insufficient for computing the off-policy policy gradient in (1), due to the emphasis term. This limitation restricts the analysis of target-based AC algorithms to the on-policy setting (Barakat et al., 2022). This motivates us to develop a functional-approximation–based formulation, under which the off-policy gradient can be computed solely from the estimation of the $Q$ function.

## 3 METHODOLOGY

In this section, we introduce our off-policy actor-critic framework. §3.1 presents a general meta algorithm based on function approximations of actor and functional critic. We then investigate the theoretical guarantees under linear functional approximation in §3.2, describing the key sub-routines in detail and discussing possible alternatives that can be integrated into the meta-algorithm. In §4, we provide a minimal neural network-based implementation and preliminary experimental results.

### 3.1 META ALGORITHM: FUNCTIONAL ACTOR CRITIC

The overall procedure of our framework is presented as a meta-algorithm composed of modular sub-routines for policy evaluation (corresponding to the functional critic update) and policy improvement (corresponding to the actor update), as summarized in Algorithm 1.

---

**Algorithm 1** Functional Actor-Critic Meta Algorithm

1: **Inputs:** Initial actor parameter $\theta_0$, initial functional critic parameter $\xi_0$, number of epochs $T$, batch size $m$, actor update step-size schedule $\{\eta_t\}_{t=1}^T$.
2: **for** $t = 1, \ldots, T$. **do**
3:     Sample $\mathcal{D}_t \leftarrow (s_t, a_t, r_t, s'_t)$ from the data-collection policy.
4:     Update functional critic parameter $\xi_t \leftarrow$**Functional Policy Evaluator** $(t, \xi_{t-1}, \theta_{t-1}, \mathcal{D}_t)$
5:     Compute the **Parameterized Off-Policy Gradient** $G_t \leftarrow \nabla_\theta\big(\mathcal{J}(\pi_\theta; \xi_t)\big)|_{\theta=\theta_{t-1}}$ // See (4)
6:     Update the actor parameter $\theta_t \leftarrow \theta_{t-1} - \eta_t G_t$. //Gradient Update
7: **end for**
8: **Return** $\theta_T$.

---

**Functional Policy Evaluator.** The critic is defined as a trainable functional $\hat{Q}(\cdot; \xi)$, parameterized by $\xi$, which takes as input a triplet $(\pi_\theta, s, a) \in \Theta \times \mathcal{S} \times \mathcal{A}$ and aims to approximate the policy value

$Q_\theta(s, a)$. This further gives the functional value evaluator

$$\mathcal{J}(\pi_\theta; \xi) := \sum_{a,s} \pi_\theta(a|s) d_\mu(s) \hat{Q}(\pi_\theta, s, a; \xi).$$

We emphasize a key motivation behind our functional evaluator: to decouple policy optimization from value estimation while enabling the critic to generalize across time-varying input policies. Unlike standard actor-critic methods that rely on a two-timescale schedule to chase a "moving target"—where the critic must be updated many times for each actor update, and then re-updated after each actor change—our functional critic is designed to accumulate knowledge across policies in an explicit manner. The idea behind this design is that the value functions of continually changing policies often share common structures that can be effectively captured by sufficiently expressive functionals—such as those parameterized by large-scale neural networks.

Following this insight, we formulate the policy evaluation of changing policies as a continual TD-learning problem over *functionals*, where the learning objective at time step $t$ is to match the Bellman equation for the policy $\pi_{\theta_t}$ at that time step,

$$\hat{Q}(\pi_{\theta_t}, s, a; \xi) \approx r_{\pi_{\theta_t}}(s, a) + \gamma \mathbb{E}_{s'}[\sum_{a'} \pi_{\theta_t}(a'|s') \hat{Q}(\pi_{\theta_t}, s', a'; \xi)|s, a], \tag{3}$$

whereas the expectation over $s'$ can be estimated from data. This formulation enables the functional critic to adapt continuously to time-varying policy inputs without requiring reset or retraining after each policy update—eliminating the need for two-timescale schedules, which often slow down the policy learning, while maintaining training stability.

**Parametrized Off-Policy Gradient.**   The actor update sub-routine in our framework follows the policy gradient paradigm. However, a key distinction lies in how we compute the exact off-policy gradient: it can be derived directly from the learned functional critic $\hat{Q}(\pi_{\theta_t}, s, a; \xi)$,

$$\nabla_\theta \mathcal{J}(\pi_\theta; \xi) = \mathbb{E}_{s \sim d_\mu} \big[ \sum_{a \in \mathcal{A}} \hat{Q}(\pi_\theta, s, a; \xi) \nabla_\theta \pi_\theta(a|s) + \pi_\theta(a|s) \nabla_\theta \hat{Q}(\pi_\theta, s, a; \xi)) \big]. \tag{4}$$

Compared with prior approaches, as discussed in §2 around (1), our formulation neither drops the second term as most empirical approaches, which introduces additional errors, nor relies on emphasis estimation as in Imani et al. (2018); Zhang et al. (2020), which brings further instability challenges. The simplicity of this exact off-policy gradient formula, again, shows the power of our proposed functional critic modeling.

### 3.2 THEORETICAL BENEFITS: A CASE STUDY UNDER LINEAR FUNCTIONAL APPROXIMATION

In this section, we present a theoretical analysis of the functional actor-critic framework to illustrate its benefits. We focus on the *linear functional approximation* setting—an analogy of the widely studied linear function approximation setting that serves as a foundational step toward understanding more complex scenarios (Zhang et al., 2020; Zhang, 2022; Zhang and Whiteson, 2022; Sutton et al., 2009). More precisely, we impose the following structural assumption to design tractable functional critics update algorithms with provable convergence guarantees:

**Assumption 1.** *Given a policy class $\Pi := \{\pi_\theta\}_{\theta \in \Theta}$ parameterized by $\Theta$, there exists a known feature map $\phi : \mathcal{S} \times \mathcal{A} \times \Theta \to \mathbb{R}^d$ and underlying $\xi_0 \in \mathbb{R}^d$ so that*

$$\max_{\theta, s, a} |Q_\theta(s, a) - \phi(s, a; \theta)^\top \xi_0| \le \epsilon$$

**Remark 1.** *It is worth noting that with perfect knowledge, an exact linear functional representation always exists even in the case $d = 1$. Specifically, under the scalar parameterization $\phi(s, a; \theta) = Q_\theta(s, a)$, the choice $\xi_0 = 1$ yields $Q_\theta(s, a) = \phi(s, a; \theta)^\top \xi_0$ for all $s, a, \theta$. This illustrates the improved expressive power when parameter-specific feature maps are permitted. Throughout this section, we assume a known feature map for simplicity of theoretical analysis, consistent with most prior work on linear function approximation (Zhang et al., 2020; Zhang, 2022; Zhang and Whiteson, 2022; Sutton et al., 2009). In practice, however, the policy-dependent feature map $\phi(\cdot)$ must be learned via a dedicated subroutine (see §4 for an exemplar neural network-based implementation), and $\epsilon$ captures the error introduced by the imperfect learning of this feature map. We also provide a explanation based on local expansion in Appendix A.2 to provide more motivations on Assumption 1.*

We further impose the following regularity assumptions on the feature map and policy, which is standard for AC analysis (Konda, 2002; Sutton et al., 2009; Zhang et al., 2020; Barakat et al., 2022).

**Assumption 2.** *There exists a constant $C_0 < \infty$ such that for all $(s, a, \theta, \theta')$,*

$$\max\left\{ \|\phi(s, a; \theta)\|, \|\nabla\phi(s, a; \theta)\|\}, \frac{|\pi_\theta(a|s) - \pi_{\bar\theta}(a|s)|}{\|\theta - \theta'\|}, \frac{\|\phi(s, a; \theta) - \phi(s, a; \theta')\|}{\|\theta - \theta'\|} \right\} \le C_0$$

**Functional Critic Design with Target Critic.** Under Assumption 1, we design the Functional Policy Evaluator sub-routine (line 4 of Algorithm 1) using a target functional critic, as illustrated in Algorithm 2, consistent with the strategy employed by most empirically successful AC methods (Fujimoto et al., 2018; Lillicrap et al., 2015; Haarnoja et al., 2018). For a gradient TD-based approach (Sutton et al., 2009; Yu, 2017; Zhang et al., 2020), which has appeared frequently in previous theoretical work but lacks empirical success, we include the details in Appendix C.

In Algorithm 2, critic parameters $\xi_t$ and target parameters $w_t$ are updated simultaneously, where linear forms $\hat{Q}(\pi_\theta, s, a; \xi) = \phi(s, a; \theta)^\top \xi$, $\hat{Q}_{\text{tar}}(\pi_\theta, s, a; w) = \phi(s, a; \theta)^\top w$ are applied for approximating $Q_\theta(s, a)$. At each time-step of Algorithm 2, the target variable $w_{t-1}$ is applied to produce the predicted value $\hat{V}_{\text{tar}}(\pi_\theta, s'_t; w_{t-1}) := \sum_a \pi_\theta(a|s'_t)\hat{Q}_{\text{tar}}(\pi_\theta, s'_t, a; w_{t-1})$, then the critic variable is updated as line 2 via $\lambda$-regularized on-policy TD under the target value prediction. On the other hand, the target variable $w_t$ is updated from the (projected) linear residual with $\xi_t$, where the projection operators $\Gamma_1, \Gamma_2$ are applied for technical reasons (Zhang et al., 2021).

---

**Algorithm 2** Functional Policy Evaluator — Target Based Linear Critic Update

1: **Inputs:** global time-step $t$, actor parameter $\theta$, critic parameter $\xi_{t-1}$, input data $(s_t, a_t, r_t, s'_t)$
2: **Initialize:** Global target functional parameter $w_0$(only when $t = 1$), critic update schedule $\{\alpha_t, \beta_t\}_{t=1}^{+\infty}$. Truncation levels $B_1, B_2$ and corresponding truncation functions $\Gamma_i(z) := B_i \cdot z/\|z\|_2$. Regularization factor $\lambda$
3: $\xi_t \leftarrow (1 - \alpha_t\lambda)\xi_{t-1} + \alpha_t\left(r_t + \gamma\left[\sum_a \pi_\theta(a|s'_t)\phi(s'_t, a; \theta)^\top w_{t-1} - \phi(s_t, a_t; \theta)^\top \xi_{t-1}\right)\phi(s_t, a_t; \theta)\right.$
   // Value update with target value prediction
4: $w_t \leftarrow \Gamma_1\left(w_{t-1} + \beta_t\left(\Gamma_2(\xi_t) - w_{t-1}\right)\right)$ // Target variable update with truncation
5: **Return** $\xi_t$.

---

As in previous works, we impose the following two-timescale schedule[2] between $\alpha_t$ and $\beta_t$:

**Assumption 3.** *The step-size schedule $\{(\alpha_t, \beta_t)\}_{t=1}^\infty$ satisfies*

1. ***Robbins-Monro Condition:*** $\sum_{t=1}^\infty \alpha_t = \sum_{t=1}^\infty \beta_t = +\infty, \sum_{t=1}^\infty \alpha_t^2 + \beta_t^2 < +\infty.$

2. ***Two-Timescale Condition:*** $\exists\gamma > 0$ such that $\sum_{t=1}^\infty (\beta_t/\alpha_t)^\gamma < +\infty.$

We now present a heuristic analysis of the dynamics of $(\xi_t, w_t)$ under Assumption 3 in the fixed-$\theta$ setting, to build intuition for our policy evaluation guarantee. A formal result for the time-varying case $\theta_t$ is given in Theorem 1. Under Assumption 3 and given large enough $\lambda$, we have the dynamic of $\xi_t$ and $w_t$ can be governed by the continuous time dynamics $\bar\xi(t), \bar{w}(t)$ satisfying that

$$\textbf{fast timescale dynamic: } \|\bar\xi(t) - \bar{G}(\theta)^{-1}\bar{h}(\theta, \bar{w}(t))\| = 0$$

$$\textbf{slow timescale dynamic: } \frac{d}{dt}\bar{w}(t) = \bar\xi(t) - \bar{w}(t) \tag{5}$$

For $\bar{G}(\theta) := \mathbb{E}_{(s,a)\sim d_\mu}\left[\phi(s, a; \theta)\phi(s, a; \theta)^\top\right] + \lambda I$ and

$$\bar{h}(\theta, w) := \mathbb{E}_{(s,a)\sim d_\mu, s'\sim P(\cdot|s,a)}\left[\left(r(s, a) + \gamma\sum_{a'}\phi(s', a'; \theta)^\top w\right)\phi(s, a; \theta)\right].$$

---

[2]We note that the two-timescale schedule introduced here is distinct from the one mentioned in most part of this paper—typically used between actor and target critic (or critic in GTD methods) updates in AC. Our schedule here is an outcome of target critic and is applied solely for evaluation purposes, since the convergence of critic performance essentially depends on the convergence of target variable $w_t$.

Ideally, this system should converge to the stationary solution satisfying $\frac{d}{dt}\bar{w}(t) = 0$, which is given by the $\lambda$-regularized LSTD solution,

$$w_\lambda^\star(\theta) = \text{argmin}_w \mathbb{E}_{d_\mu}\left[\left(r(s,a) + \gamma \sum_{a'} \pi(a'|s')\phi(s',a';\theta)^\top w - \phi(s,a;\theta)^\top w\right)^2\right] + \lambda\|w\|^2, \quad (6)$$

which then gives a desired critic evaluation guarantee under fixed $\theta$.

From the this informal analysis, we see that the role of the critic variable $\xi$ is to rapidly converge to a function determined by the target variable $w$. The convergence of $w$ then leads to the convergence of the entire evaluation procedure, consistent with our earlier statement that the target critic variable drives the overall critic convergence process.

To rigorously extend the above intuition to time-varying $\theta_t$, we impose the following assumptions:

**Assumption 4.** *The chain in $\mathcal{S} \times \mathcal{A}$ induced by $\mu$ is ergodic.*

**Assumption 5.** *There exists constants $c, C > 0$ so that for any given $\theta$, $\|\phi(s,a;\theta)\|_2 \leq C$ and the matrices*

$$\mathbb{E}_{(s,a)}[\phi(s,a;\theta)\phi(s,a;\theta)^\top] \succeq cI. \quad (7)$$

**Assumption 6.** *There exists some $\Delta_\lambda$ depending on $\lambda$ so that for all $\theta, \theta'$, we have*

$$\|w_\lambda^\star(\theta) - w_\lambda^\star(\theta')\| \leq \Delta_\lambda \|\theta - \theta'\| \quad (8)$$

The Assumption 4 is standard for off-policy RL literature (Zhang et al., 2020; 2021), while the Assumption 5 can be seen as an analogue of those in Sutton et al. (2009); Zhang et al. (2020; 2021). Finally, since different policies $\pi_\theta$ correspond to different $\lambda$-LSTD solutions, we use Assumption 6 to characterize the heterogeneity of the $\lambda$-regularized LSTD solutions across $\theta$. While Assumption 6 always holds under Assumptions 2, 5, and (6), the value of $\Delta_\lambda$ can be significantly smaller under the functional representation Assumption 1. For example, in the ideal setting where a perfect linear functional $w^\star$ exists such that $\phi(s,a;\theta)^\top w^\star = Q_\theta(s,a)$, Assumption 6 is satisfied with $\Delta_\lambda = 0$.

Under these assumptions, we have the following guarantee of policy evaluation.

**Theorem 1.** *Suppose Assumption 2–6 holds and $\limsup_t \eta_t/\beta_t \leq \kappa$. Then for $B_1 > B_2 > C, \lambda \geq \max\{4\gamma^2 C^2, 4C/B_1\}$, the $\xi_t$ updated in Algorithm 2 satisfies*

$$\limsup_t [\|\xi_t - w_\lambda^\star(\theta_t)\|_2^2] \lesssim \kappa\Delta_\lambda.$$

Theorem 1 establishes an approximation guarantee for $\xi_t$ with respect to the $\lambda$-regularized LSTD path associated with $\theta_t$. In particular, it no longer requires a strict two-timescale schedule between the critic step-size $\beta_t$ and the actor step-size $\eta_t$, but only an upper bound $\kappa$ on their ratio. As $\kappa \to 0$, this recovers the standard two-timescale behavior, namely $\limsup_t \|\xi_t - w_\lambda^\star(\theta_t)\|_2^2 = 0$ without any condition on $\Delta_\lambda$. A potentially more insightful observation is that when $\Delta_\lambda$ is small (for example, in the scenario described below Assumption 1), $\xi_t$ still provides a good approximation to the regularized LSTD path even for constant $\kappa$. This improves the previous three-timescale schedule needed for convergence of even on-policy target-based AC (Barakat et al., 2022).

**Remark 2** (Possible Alternative of Algorithm 2). *While we consider the target network based evaluation mainly for its good empirical performance and matching our practical algorithm design, as detailed in § 4. Alternative approaches such as GTD2 also works in our setting, in particular, with GTD 2, we don't even to require the two-timescale between target and critic parameters, we leave the full detailed algorithm and theory to Appendix C for completeness.*

With Theorem 1, we obtain the following AC convergence guarantee by directly converting the gradient estimation error into the dynamics of $\theta_t$. This follows a standard technique from Zhang et al. (2020); Konda (2002), so we omit the proof. To the best of our knowledge, Theorem 2 provides the first convergence result for an off-policy target-based AC algorithm under function approximation.

**Theorem 2** (Convergence of Algorithm 1). *Under the same conditions as in Theorem 1, and introduce the gradient bias term due to linear functional approximation*

$$b(\theta_t) := \nabla J(\theta_t) - \left[\sum_a \phi(s,a;\theta_t)^\top w_\lambda^\star(\theta_t)\nabla_\theta \pi_{\theta_t}(a|s) + \pi_{\theta_t}(a|s)\nabla_\theta \phi(s,a;\theta_t)^\top w_\lambda^\star(\theta_t)\right]$$

*the iterates $\{\theta_t\}$ generated by Functional AC (Algorithm 1) satisfy*

$$\liminf_{t \to \infty} \left( \|\nabla J(\theta_t)\| - \|b(\theta_t)\| \right) \lesssim \kappa \Delta \quad \text{almost surely.}$$

*That is, $\{\theta_t\}$ visits any neighborhood of the set $\{\theta : \|\nabla J(\theta)\| \lesssim \|b(\theta)\| + \kappa \Delta\}$ infinitely often.*

## 4 PRACTICAL IMPLEMENTATION

For real-world applications, instead of the linear functional approximation described in §3.2, we use modern neural networks to parameterize functional critics, target functional critics, and actors.

**Functional critics and target critics.** We implement an ensemble of functional critics $\{\hat{Q}^{(i)}\}_{i=1}^n$, each of which consists of three encoders, a transformer-based actor encoder $E_{act}^{(i)}$ with parameters $\xi_{act}^{(i)}$, a MLP-based state-action encoder $E_{sa}^{(i)}$ with parameters $\xi_{sa}^{(i)}$, and a MLP-based joint encoder $E_{joint}^{(i)}$ with parameters $\xi_{joint}^{(i)}$. Outputs of $E_{act}^{(i)}$ and $E_{sa}^{(i)}$ are concatenated and feeded into the joint encoder $E_{joint}^{(i)}$ to get an estimate of the state-action value for the input policy, i.e.,

$$\hat{Q}^{(i)}(\pi_{\theta_t}, s, a; \xi_{act}^{(i)}, \xi_{sa}^{(i)}, \xi_{joint}^{(i)}) = E_{joint}^{(i)}\left( E_{act}^{(i)}(\pi_{\theta_t}; \xi_{act}^{(i)}), E_{sa}^{(i)}(s, a; \xi_{sa}^{(i)}); \xi_{joint}^{(i)} \right). \qquad (9)$$

For target functional critics, we only maintain delayed update copies of the state-action encoder $E_{sa}$ and the joint encoder $E_{joint}$, while sharing the same actor encoder $E_{act}$ with the corresponding functional critic, i.e.,

$$\hat{Q}_{tar}^{(i)}(\pi_{\theta_t}, s, a; \xi_{act}^{(i)}, \xi_{sa}^{'(i)}, \xi_{joint}^{'(i)}) = E_{joint}^{(i)}\left( E_{act}^{(i)}(\pi_{\theta_t}; \xi_{act}^{(i)}), E_{sa}^{(i)}(s, a; \xi_{sa}^{'(i)}); \xi_{joint}^{'(i)} \right). \qquad (10)$$

For each $\pi_{\theta_t}$ and sampled transition $(s_t, a_t, r_t, s'_t)$, following the Bellman equation (3) for $\pi_{\theta_t}$, the TD target is computed by,

$$y_t^{(i)} = r_t + \gamma \hat{Q}_{tar}^{(i)}\left( \pi_{\theta_t}, s'_t, \pi_{\theta_t}(s'_t); \xi_{act}^{(i)}, \xi_{sa}^{'(i)}, \xi_{joint}^{'(i)} \right). \qquad (11)$$

The functional critic learning minimizes the following loss,

$$L_{\hat{Q}^{(i)}}\left( \xi_{act}^{(i)}, \xi_{sa}^{(i)}, \xi_{joint}^{(i)} | (\pi_{\theta_t}, s_t, a_t, r_t, s'_t) \right) = \left( y_t^{(i)} - \hat{Q}^{(i)}(\pi_{\theta_t}, s_t, a_t; \xi_{act}^{(i)}, \xi_{sa}^{(i)}, \xi_{joint}^{(i)}) \right)^2, \forall i. \qquad (12)$$

Given a training set of actors $\mathcal{A}_t = \left\{ \pi_{\theta_t^{(j)}} \right\}$ and a sampled transition batch $\mathcal{D}_t = \{(s_t, a_t, r_t, s'_t)\}$, each functional critic is trained with $\mathcal{A}_t \times \mathcal{D}_t$ using a batched form of (12).

It is worth noting that each functional critic is trained independently without employing the default "minimum target value" heuristic commonly used in existing methods to mitigate value overestimation. This shows the effectiveness of our approach in stabilizing the off-policy TD learning.

**Actor encoders.** The design of the actor encoder $E_{act}$ has to achieve a balance between effective actor representation and computational efficiency. We use a set of trainable evaluation samples (states) $\{\zeta_i\}_{i=1}^n$ to forward the input actor, and extract the output action as well as some optional layers of hidden neurons as a sequence to feed into a transformer encoder. This idea was motivated by some functional analysis, a scalar evaluation function for actors is a mapping from $\Pi \to \mathbb{R}$, where $\Pi$ is the function space of actors. Then the simplest evaluation function is the delta function $\delta_x$ defined by $\delta_x(\pi) = \pi(x), \forall \pi \in \Pi$. So the set of evaluation samples can be viewed as a set of evaluation functions parameterized by $\{\zeta_i\}_{i=1}^n$. In this sense, $\{\zeta_i\}_{i=1}^n$ are part of the trainable parameters $\xi_{act}$ of $E_{act}$ and are trained together with other parameters of the network $\hat{Q}$ using (12).

**Deterministic actors.** Most existing AC methods depend on stochastic actors and entropy regularization to balance exploration and exploitation, but this design introduces additional hyperparameters that are often difficult to tune in practice. Given the power of our functional critic modeling, every functional critic is trained with all actors available, and can be used to calculate the off-policy gradient (4) for any actor. This unmatched property motivates us to use an ensemble $\left\{ \pi_{\theta_t^{(i)}} \right\}$ of simpler deterministic actors to achieve efficient actor learning and exploration. The actor ensemble naturally serves as training actor set $\mathcal{A}_t$ during functional critic learning. In actor learning, we pair

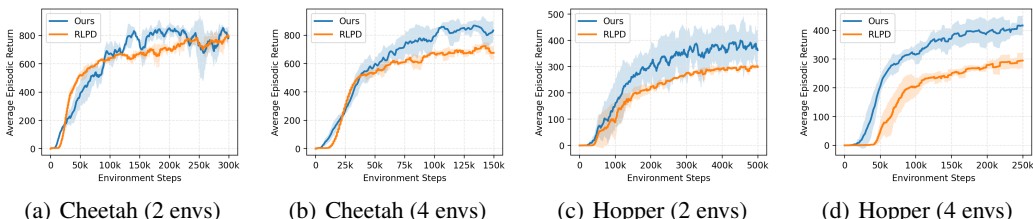

(a) Cheetah (2 envs)  (b) Cheetah (4 envs)  (c) Hopper (2 envs)  (d) Hopper (4 envs)

Figure 1: Averaged episodic return against environment steps of our method vs RLPD on Cheetah-run and Hopper-hop tasks of DM Control. "n envs" means the number of parallel environments. Results are averaged over four runs of different random seeds, with the shaded area corresponding to the standard deviation.

each actor $\pi_{\theta_t^{(i)}}$ with an individual critic $\hat{Q}^{(i)}$ and fix the pairing throughout the training procedure to achieve a Thompson sampling flavor of design for exploration. Then actor $\pi_{\theta_t^{(i)}}$ can be trained with exact off-policy gradients (4) using $\hat{Q}^{(i)}$ and sampled experience batch $\{(s_t, a_t)\}$.

Algorithm 3 in Appendix B summarizes this minimal implementation of the proposed off-policy AC framework, incorporating only the theoretical insights while avoiding additional heuristic components.

### 4.1 EMPIRICAL RESULTS

We conduct preliminary experiments comparing our implementation described in §4 with the state-of-the-art RLPD (Ball et al., 2023) algorithm on the Cheetah-run and Hopper-hop continuous control tasks of the DeepMind Control Suite (Tunyasuvunakool et al., 2020). While RLPD was originally proposed for off-policy learning with both online and offline data, it remains one of the state-of-the-art off-policy AC algorithms when applied to online data alone and is widely used as a backbone method in recent literature (Zhou et al., 2024; Xiao et al., 2025). It incorporates the most critical improvements over classic off-policy methods such as SAC and TD3—namely, a higher UTD ratio and regularization techniques (e.g. critic ensemble and layer normalization for critics) to stabilize the high UTD training—which have been extensively validated by the RL community.

We use the RLPD implementation from an actively maintained and extensively tested open-sourced RL library (Xu et al., 2021), and we implement our algorithm within the same framework to enable a controlled and fair comparison. Since our method is fundamentally different from existing AC methods in its design, we carefully align configurable hyperparameters to ensure fairness. For example, both methods employ 10 critics and target critics, and their actor and critic networks share the same architecture, aside from unavoidable differences. Specifically, our functional critic includes an additional actor encoder module, whereas the RLPD actor network incorporates extra Gaussian projection layers for stochastic outputs. Further implementation details are included in Appendix B.

The results are reported in Figure 1. Although our experiments are still preliminary, we observe several encouraging signs of the real-world effectiveness of our approach. First, it achieves performance favorable to representative state-of-the-art method without relying on widely adopted heuristics considered essential for existing off-policy AC methods. Second, while RLPD typically requires a high UTD ratio (In our experiments, 10 critic updates per actor update after grid search), our approach performs well with a critic-to-actor update ratio of just 2 or 3. We fix this ratio at 3 across all experiments. These findings support our motivation and theoretical insights that functional critic modeling effectively addresses the two major challenges of off-policy AC, enabling stable critic training without multi-timescale actor-critic updates and achieving more efficient policy improvement. Notably, we have not achieved stable training at 1:1 actor-critic update ratio. We attribute this to our relatively simple actor encoder design, which only extracts neurons from the last two layers of the actor network to form the transformer encoder's input sequence—a choice made mainly for efficiency. As a result, the generalization potential of the functional critic in the input actor space has not been fully realized. Last but not least, our method better leverages the benefits of parallel environments: the performance gap over RLPD consistently widens as the number of environments increases from two to four. We attribute this to the data-driven generalization ability of functional critic modeling, which suggests even greater potential if combined with more sophisticated exploration strategies and more aggressive parallelization.

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

**Claim of LLM Usage** The authors used Large Language Models (LLMs) to correct grammatical errors and polish sentences for improved clarity.

# A DETAILS OF SECTION 3.2

## A.1 PROOF OF THEOREM 1

To prove Theorem 1, we first analyze rigorously the two-timescale dynamic by generalzing the following result from Konda (2002):

**Theorem 3** (Konda (2002)). *Let $\{Y_t\}$ be a Markov chain with a finite state space $\mathcal{Y}$ and transition kernel $P_{\mathcal{Y}} \in \mathbb{R}^{|\mathcal{Y}| \times |\mathcal{Y}|}$. Consider an iterate sequence $\{\xi_t\} \subset \mathbb{R}^n$ evolving according to*

$$\xi_{t+1} = \xi_t + \alpha_t \left( h(Y_t; w_t) - G(Y_t; w_t)\xi_t \right),$$

*where $\{w_t\} \subset \mathbb{R}^m$ is another iterate, $h(\cdot; w) : \mathcal{Y} \to \mathbb{R}^n$ and $G(\cdot; w) : \mathcal{Y} \to \mathbb{R}^{n \times n}$ are vector-valued and matrix-valued functions parameterized by $w$, and $\alpha_t > 0$ is a step size. Suppose **certain conditions** holds, then the iterates satisfy*

$$\sup_t \|\xi_t\| < \infty, \qquad \lim_{t \to \infty} \|\bar{G}(w_t)\xi_t - \bar{h}(w_t)\| = 0 \quad a.s. \tag{13}$$

Previous results for two-timescale analysis lies in checking the "certain conditions" are satisfied so that (14) holds (Zhang et al., 2020; 2021; Barakat et al., 2022). However, the Theorem 3 cannot be applied to our setting since in our AC design, we have additional policy parameter $\theta_t$ changing every time-step, this makes Theorem 3 with fixed function $h(\cdot; \cdot)$, $G(\cdot; \cdot)$ in-applicable. Now we introduce the following generalization of Theorem 3:

**Theorem 4** (Time-changing variant of Theorem 3). *Let $\{Y_t\}$ be a Markov chain with a finite state space $\mathcal{Y}$ and transition kernel $P_{\mathcal{Y}} \in \mathbb{R}^{|\mathcal{Y}| \times |\mathcal{Y}|}$. Consider an iterate sequence $\{\xi_t\} \subset \mathbb{R}^n$ evolving according to*

$$\xi_{t+1} = \xi_t + \alpha_t \left( h_t(Y_t; w_t) - G_t(Y_t; w_t)\xi_t \right),$$

*where $\{w_t\} \subset \mathbb{R}^m$ is another iterate, $h_t(\cdot; w) : \mathcal{Y} \to \mathbb{R}^n$ and $G_t(\cdot; w) : \mathcal{Y} \to \mathbb{R}^{n \times n}$ are sequences of vector-valued and matrix-valued functions parameterized by $w$, and $\alpha_t > 0$ is a step size. Suppose Assumption 7 holds, then the iterates satisfy*

$$\sup_t \|\xi_t\| < \infty, \qquad \lim_{t \to \infty} \|\bar{G}(w_t)\xi_t - \bar{h}(w_t)\| = 0 \quad a.s. \tag{14}$$

**Assumption 7.** *Suppose the following conditions hold:*

1. *(**Stepsize**) The sequence $\{\alpha_t\}$ is deterministic, non-increasing, and satisfies the Robbins-Monro conditions:*

$$\sum_t \alpha_t = \infty, \qquad \sum_t \alpha_t^2 < \infty.$$

2. *(**Parameter Changing Rate**) The random sequence $\{w_t\}$ satisfies*

$$\|w_{t+1} - w_t\| \le \beta_t H_t,$$

*where $\{H_t\}$ is a nonnegative process with bounded moments and $\{\beta_t\}$ is a deterministic sequence such that $\sum_t \left( \frac{\beta_t}{\alpha_t} \right)^d < \infty$ for some $d > 0$.*

3. *(**Poisson Equation**) There exist functions $\hat{h}_t(\cdot; w) : \mathcal{Y} \to \mathbb{R}^n$, $\bar{h}_t(w) \in \mathbb{R}^n$, $\hat{G}_t(\cdot; w) : \mathcal{Y} \to \mathbb{R}^{n \times n}$, and $\bar{G}_t(w) \in \mathbb{R}^{n \times n}$ such that*

$$\hat{h}_t(y; w) = h_t(y; w) - \bar{h}_t(w) + \sum_{y'} P_{\mathcal{Y}}(y, y')\hat{h}_t(y'; w),$$

$$\hat{G}_t(y; w) = G_t(y; w) - \bar{G}_t(w) + \sum_{y'} P_{\mathcal{Y}}(y, y')\hat{G}_t(y'; w).$$

4. **(Boundedness)** *There exists a constant $C_0 < \infty$ such that for all $t$, $w$, and $y$,*

$$\max\left(\|\bar{h}_t(w)\|, \|\bar{G}_t(w)\|, \|\hat{h}_t(y; w)\|, \|h(y; w)\|, \|\hat{G}_t(y; w)\|, \|G(y; w)\|\right) \le C_0.$$

5. **(Lipschitz Continuity)** *There exists a constant $C_0 < \infty$ such that for all $t$, $w$, $\bar{w}$, and $y$,*

$$\max\left(\|\bar{h}_t(w) - \bar{h}_t(\bar{w})\|, \|\bar{G}_t(w) - \bar{G}_t(\bar{w})\|\right) \le C_0\|w - \bar{w}\|,$$
$$\|f(y; w) - f(y; \bar{w})\| \le C_0\|w - \bar{w}\|,$$

*where $f$ represents any of $\hat{h}_t, h_t, \hat{G}_t, G_t$.*

6. **(Uniformly Positive Definite)** *There exists $\eta_0 > 0$ such that for all $t$, $w$, and $\xi$,*

$$\xi^\top \bar{G}_t(w)\xi \ge \eta_0\|\xi\|^2.$$

Noticing that the dynamic of $\xi_t$ in Algorithm 2 corresponds to the following selection of $h_t, G_t$ :

$$\mathcal{Y} := \mathcal{S} \times \mathcal{A} \times \mathcal{S}, \quad y_t = (s_t, a_t, s_t')$$

$$h_t(s, a, s'; w) := \left[r(s, a) + \gamma \sum_{a'} \pi_{\theta_t}(a'|s')\phi(s', a'; \theta_t)^\top w\right]\phi(s, a; \theta_t),$$

$$G_t(s, a, s'; w) := \phi(s, a; \theta_t)\phi(s, a; \theta_t)^\top + \eta I.$$

Now we verify the conditions in Assumption 7:

*Verification of Assumption 7.* Condition 1 and 2 in Assumption 7 are satisfied by our assumption 3. Condition 3 is satisfied with the selection

$$\bar{h}_t(w) := \mathbb{E}_{(s,a) \sim d_\mu, s' \sim P(\cdot|s,a)}[h_t(s, a, s'; w)],$$
$$\bar{G}_t(w) := \mathbb{E}_{(s,a) \sim d_\mu, s' \sim P(\cdot|s,a)}[G_t(s, a, s'; w)],$$

and

$$\hat{h}_t(s, a, s'; w) := \mathbb{E}[\sum_{t=0}^\infty h_t(s, a, s'; w) - \bar{h}_t(w)|s_0 = s, a_0 = a, s_0' = s'],$$

$$\hat{G}_t(s, a, s'; w) := \mathbb{E}[\sum_{t=0}^\infty G_t(s, a, s'; w) - \bar{G}_t(w)|s_0 = s, a_0 = a, s_0' = s'].$$

Condition 4,5,6 are satisfied by our assumption 5 and the form of $h_t, G_t$. $\qquad\square$

With Theorem 4, we have then

$$\lim_t \|\xi_t - \bar{G}_t(w_t)^{-1}\bar{h}_t(w_t)\| = 0 \tag{15}$$

almost surely. In the following, we abuse the notation

$$\bar{G}(\theta_t) := \bar{G}_t(w)(\text{since } \bar{G}_t(w) \text{ is independent of } w), \bar{h}_t(w) = \bar{h}(\theta_t, w)$$

to present them as a explicit function of $\theta_t, w$ for clarity.

On the other hand, with condition 15, as shown in the proof of Theorem 1 in Zhang et al. (2021), we have the dynamic of $w_t$, given by the line 4 of Algorithm 2:

$$w_t \leftarrow \Gamma_1\big(w_{t-1} + \beta_t\big(\Gamma_2(\xi_t) - w_{t-1}\big)\big) \tag{16}$$

is asymptotically equivalent to

$$w_t \leftarrow w_{t-1} + \beta(\bar{G}(\theta_t)^{-1}\bar{h}(\theta_t, w_{t-1}) - w_{t-1}) \tag{17}$$

as long as we can show the following condition:

$$\sup_{\theta, \|w\| \le B_1} \bar{G}(\theta)^{-1}\bar{h}(\theta, w) < B_2 < B_1 \tag{18}$$

holds.

Now we show (18) can hold with our selection of regularization factor $\eta$:

*Proof of* (18). First noticing that for any $w, w', \theta$, we have as shown in (20) of Zhang et al. (2021)

$$\|\bar{G}(\theta)^{-1}\big[\bar{h}(\theta, w) - \bar{h}(\theta, w')\big]\| \leq \frac{\gamma C}{2\sqrt{\eta}}\|w - w'\|,$$

thus as long $\eta > 4\gamma^2 C^2$, we have

$$\|\bar{G}(\theta)^{-1}\big[\bar{h}(\theta, w) - \bar{h}(\theta, w')\big]\| \leq \frac{1}{4}\|w - w'\|. \tag{19}$$

Now for any $w$ with $\|w\|_2 \leq B$, by set $w' = 0$, we get

$$\|\bar{G}(\theta)^{-1}\bar{h}(\theta, w) - \bar{G}(\theta)^{-1}\mathbb{E}_{(s,a)\sim d_\mu}[r(s,a)\phi(s,a;\theta)]\| \leq \frac{B_1}{2}$$

$$\implies \|\bar{G}(\theta)^{-1}\bar{h}(\theta, w)\| \leq \frac{B_1}{4} + \eta^{-1}C.$$

Thus as long as $\eta \geq \max\{\gamma^2 C^2, 4C/B_1\}$, we have then

$$\sup_{\|w\| \leq B_1, \theta} \|\bar{G}(\theta)^{-1}\bar{h}(\theta, w)\| \leq \frac{3}{4}B_1,$$

as desired. $\qquad\square$

Now it suffice to analysis (17): For any $\theta$, denoting $w^\star(\theta)$ as the fixed point satisfying

$$w^\star(\theta_t) = \bar{G}(\theta)^{-1}\bar{h}(\theta, w),$$

which is unique by (19). Now the dynamic of (17) can be further decomposed as

$$w_t - w^\star(\theta_t) = w_{t-1} - w^\star(\theta_{t-1}) + \beta_t\big[\bar{G}(\theta_t)^{-1}\bar{h}(\theta_t, w_{t-1}) - w_{t-1}\big] + \underbrace{w^\star(\theta_t) - w^\star(\theta_{t-1})}_{=O(\kappa\beta_t\Delta)}$$

$$= w_{t-1} - w^\star(\theta_{t-1}) + \beta_t\big[\bar{G}(\theta_t)^{-1}\bar{h}(\theta_t, w_{t-1}) - w_{t-1} + O(\kappa\Delta)\big].$$

Then the behaviour of $w_t - w^\star(\theta_t)$ can be reduced to the continuous-time dynamic for $z(t) := w(t) - w^\star(\theta(t))$ : We have

$$\frac{d}{dt}z(t) = \bar{G}(\theta(t))^{-1}\bar{h}(\theta(t), w(t)) - w(t) + O(\kappa\Delta).$$

As a result, we have

$$\frac{d}{dt}\|z(t)\|^2 = 2\langle\bar{G}(\theta(t))^{-1}\bar{h}(\theta(t), w(t)) - w(t) + O(\kappa\Delta), z(t)\rangle$$

$$= 2\langle\bar{G}(\theta(t))^{-1}\bar{h}(\theta(t), w(t)) - w^\star(\theta(t)) - z(t), z(t)\rangle + O(\kappa\Delta\|z(t)\|)$$

$$\leq -\frac{1}{2}\|z(t)\|^2 + O(\kappa\Delta\|z(t)\|).$$

where the last line is by (19). As a result, we get

$$\limsup_t\|z(t)\| = O(\Delta),$$

which then indicates that $\limsup_t\|w_t - w^\star(\theta_t)\| = O(\kappa\Delta)$, as desired.

## A.2 COMMENTS ON ASSUMPTION 1

In this section, we provide another perspective on understanding Assumption 1. Given a parametrized policy space $\Pi_\Theta := \{\pi_\theta : \theta \in \Theta\}$, the $Q_{(\cdot)}(s, a)$ function at each fixed $(s, a)$ pair can be understood as map from $\Pi_\Theta$ to $\mathbb{R}$. In most previous works, the continuity of $Q$ in $\theta$ is not well-explored, while it is quite natural assumption. Now if we equip $\Pi_\Theta$ with a smooth manifold structure with tangent space $T_{\pi_\theta}$ and exponential map $\exp_{\pi_\theta}(\cdot) : T_{\pi_\theta} \to \Pi_\Theta$, then given any $\pi_{\theta_0}$ and $\pi_\theta$ *near to* $\pi_{\theta_0}$, the following *linear approximation* holds near $\pi_{\theta_0}$:

$$Q_\theta(s, a) = Q_{\exp_{\pi_{\theta_0}}(\log_{\pi_{\theta_0}}(\pi_\theta))}(s, a) \approx_{\text{linear approximation}} \Phi(s, a)^\top \underbrace{w_{\pi_{\theta_0}}\log_{\pi_{\theta_0}}(\pi_\theta)}_{:=w_{\pi_{\theta_0}}(\pi_\theta)}.$$

If assuming we can access to an oracle on computing the exponential map, thus also the logarithmic map, and denoting $\phi(s, a; \theta) := \log_{\pi_{\theta_0}}(\pi_\theta)\Phi(s, a)^\top$ our linear function approximation assumption is reduced to

$$Q_\theta(s, a) \approx \phi(s, a; \theta)^\top w_{\pi_{\theta_0}} \tag{20}$$

for $\theta$ near $\theta_0$. In this sense, for any $\theta_0$, Assumption 1 holds locally for $\theta$ close to $\theta_0$, with the underlying representation depending on $\theta_0$. The error term then reflects the approximation error incurred by the local expansion around $\theta_0$.

## B IMPLEMENTATION AND EXPERIMENT DETAILS

The minimal implementation of the proposed off-policy AC framework described in §4 is summarized in Algorithm 3. Hyperparameters across all experiments reported in Figure 1 is summarize in Table 1.

---

**Algorithm 3** Neural Network-based Functional Actor-Critic Algorithm

---

1: Initialize the actor ensemble $\mathcal{A}_0$ parameters $\{\theta_0^{(i)}\}_{i=1}^n$
2: initialize functional critic ensemble parameters $\{\xi_{act}^{(i)}, \xi_{sa}^{(i)}, \xi_{joint}^{(i)}\}_{i=1}^n$
3: Initialize extra target functional critic ensemble parameters $\{\xi_{sa}^{'(i)}, \xi_{joint}^{'(i)}\}_{i=1}^n$
4: Initialize empty replay buffer $\mathcal{R}$
5: Select number of epochs $T$, transition batch size $m$ for training, functional critic UTD $G$
6: **for** $t = 1, \ldots, T$. **do**
7:     Resample rollout actor index id from $\{1, \ldots, n\}$ if starting a new episode
8:     Take action $a_t = \pi_{\theta_t^{(\mathrm{id})}}(s_t)$
9:     Store transition $(s_t, a_t, r_t, s_{t+1})$ to buffer $\mathcal{R}$
10:     **for** $g = 1, \ldots, G$. **do**
11:       Sample transition batch $B_C$ from the buffer $\mathcal{R}$
12:       **for** $i = 1, \ldots, n$. **do**
13:         Compute TD targets (11) for batch $\mathcal{A}_t \times B_C$
14:         Update $\{\xi_{act}^{(i)}, \xi_{sa}^{(i)}, \xi_{joint}^{(i)}\}$ minimizing a batched ($\mathcal{A}_t \times B_C$) version of (12)
15:       **end for**
16:       Update target critics $\xi_{sa}^{'(i)} \leftarrow \rho\xi_{sa}^{'(i)} + (1 - \rho)\xi_{sa}^{(i)}$, $\xi_{joint}^{'(i)} \leftarrow \rho\xi_{joint}^{'(i)} + (1 - \rho)\xi_{joint}^{(i)}$
17:     **end for**
18:     Sample transition batch $B_A$ from the buffer $\mathcal{R}$
19:     **for** $i = 1, \ldots, n$. **do**
20:       Evaluate exact off-policy gradient (4) with $\hat{Q}^{(i)}$ and $B_A$
21:       Update $\theta_t^{(i)}$
22:     **end for**
23: **end for**

---

## C A GRADIENT-TD BASED CRITIC AND RELATED CONVERGENCE RESULTS

In this Appendix, we present a GTD2 (Sutton et al., 2009) based algorithm design for functional critic, under access to a perfect feature map $\phi(s, a; \theta)$ so that

$$\phi(s, a; \theta)^\top w^\star = Q_\theta(s, a)$$

for all $\theta \in \Theta$. We also introduce the notation

$$\psi(s; \theta) := \sum_a \pi(a|s)\phi(s, a)$$

for convenience, under which we have $\psi(s; \theta)^\top w^\star = V_\theta(s)$ for all $\theta \in \Theta$.

Table 1: Hyperparameters used across all reported experiments.

| Hyperparameter | Value |
|---|---|
| optimizer | Adam |
| learning rate | $3 \times 10^{-4}$ |
| batch size | 256 |
| discount factor ($\gamma$) | 0.99 |
| target update rate ($\tau$) | 0.005 |
| actor network hidden layers | (256, 256) |
| critic state-action encoder hidden layers | (256, 256) |
| critic joint encoder hidden layers | (256, 256) |
| actor transformer encoder number of layers | 4 |
| actor transformer encoder number of heads | 1 |
| size of trainable evaluation samples for actor encoder | 512 |
| actor network layers to extract hidden neurons for actor encoder | last two |
| actor encoding dimension | 128 |
| state-action encoding dimension | 64 |
| activation function | ReLU |

**A GTD2 Functional Critic.** From the primal-dual derivation in Sutton et al. (2009), we adopt the following update of $U_t$ and additional argument variables $\{\nu_t\}$ :

$$\nu_{t+1} \leftarrow \nu_t + \alpha_t \big( r_t + \gamma \psi(s_t'; \theta_t)^\top \xi_t - \psi(s_t; \theta_t)^\top \xi_t - \phi(s_t; \theta_t)^\top \nu_t \big) \psi(s_t; \theta_t)$$

$$\xi_{t+1} \leftarrow \xi_t + \alpha_t (\psi(s_t; \theta_t) - \gamma \psi(s_t'; \theta_t)) \psi(s_t; \theta_t)^\top \nu_t.$$

In the following we proceed the analysis under the same assumptions as in Theorem 1, whereas Assumption 3 only imposed for $\alpha_t$, and Assumption 6 be replaced by the following assumption as in Sutton et al. (2009):

**Assumption 8.** *There exists some constant $c_0 > 0$ so that*

$$\sigma_{\min} \big( \mathbb{E}_{(s,a) \sim d_\mu} [\psi(s; \theta) \psi(s; \theta)^\top] \big) \geq c_0,$$

$$\sigma_{\min} \big( \mathbb{E}_{s,a,s'} [(\psi(s; \theta) - \gamma \psi(s'; \theta)) \psi(s; \theta)^\top] \big) \geq c_0.$$

In particular Assumption 8 implies that $w^\star$ should be the unique solution of the equation

$$\underbrace{\mathbb{E}[\psi(s; \theta_t)(\psi(s; \theta_t) - \gamma \psi(s'; \theta_t))^\top]}_{:=A(\theta_t)} w^\star = \underbrace{\mathbb{E}_s[r_{\theta_t}(s) \psi(s; \theta_t)]}_{:=z(\theta_t)}. \tag{21}$$

for all $\theta \in \Theta$.

To show that the dynamic of $\xi_t$ converges to the solution of (21), denote $\rho_t := (\nu_t, \xi_t)^\top$, then the GTD dynamic can be written as

$$\rho_{t+1} \leftarrow \rho_t - \alpha_t \underbrace{\begin{bmatrix} -\psi(s_t; \theta_t) \psi(s_t; \theta_t)^\top & -A(\theta_t) \\ A(\theta_t)^\top & 0 \end{bmatrix}}_{:=G(\theta_t)} \rho_t + \underbrace{\begin{bmatrix} z(\theta_t) \\ 0 \end{bmatrix}}_{:=g(\theta_t)}.$$

In particular, by Assumption 8, we have

$$G(\theta) \begin{bmatrix} \nu \\ \xi \end{bmatrix} + g(\theta) = 0 \iff A(\theta) \xi = z(\theta) \iff \xi = w^\star.$$

Thus it suffices to show that the dynamic of $\rho_t$ satisfies

$$\lim_t \| G(\theta_t) \rho_t + g(\theta_t) \| = 0$$

For this purpose, we show the following time-varying version of the stochastic approximation conditions in Borkar and Meyn (2000) hold. For the function $h(\rho; \theta_t) := G(\theta_t) \rho + g(\theta_t)$ and

$$M_t := \big( \psi(s_t; \theta_t)(\psi(s_t; \theta_t) - \gamma \psi_t(s_t'; \theta_t))^\top - G(\theta_t) \big) \rho + \big( r(s_t; \theta_t) \psi(s_t; \theta_t) - g(\theta_t) \big),$$

we can verify that:

1. The function $h(\cdot; \theta) : \mathbb{R}^d \to \mathbb{R}^d$ is Lipschitz continuous, and the limit

$$h_\infty(\rho; \theta_t) := \lim_{r \to \infty} \frac{h(r\rho; \theta_t)}{r}$$

   is well-defined for every $\rho, \theta$ by Assumption 2.

2. (a) The sequence $\{(M_k, \mathcal{F}_k)\}$ is a *martingale difference sequence*, i.e.,

$$\mathbb{E}[M_{k+1} \mid \mathcal{F}_k] = 0 \quad \text{a.s.}$$

   (b) There exists a constant $c_0 > 0$ such that

$$\mathbb{E}\left[\|M_{k+1}\|^2 \mid \mathcal{F}_k\right] \leq c_0 \left(1 + \|\rho_k\|^2\right), \quad \text{a.s.}$$

   for any initial parameter vector $\rho_1 \in \mathbb{R}^{2d}$ by Assumption 5.

3. The step-size sequence $\{\alpha_t\}$ satisfies the RM conditions by Assumption 3

4. For every $\theta \in \Theta$, the limiting ODE

$$\dot{\rho}_t = h_\infty(\rho_t; \theta)$$

   has the origin as a *globally asymptotically stable equilibrium* by Assumption 8.

5. The ODE
$$\dot{\rho} = h(\rho)$$

   has a *unique globally asymptotically stable equilibrium* by Assumption 8.

Within all above conditions, we have then

$$\lim_t \|G(\theta_t)\rho_t - h(\theta_t)\|_2 \to 0.$$

This gives the desired claim. $\qquad\square$

**Remark 3.** *Unlike the regularization-based approach we used in Section 3.2, in this section we assume a stronger non-singular condition 8 as in Sutton et al. (2009) and assumes a information of perfect feature maps without representation error. Under such stronger assumptions, we show that the off-policy evaluation dynamic of GTD 2 can be fully decoupled with the $\eta$ update dynamic. On the other hand, when we only have Assumption 5, a $\lambda$-regularization based approach as in Zhang et al. (2020) should be applied to obtain a $\Delta_\lambda$-dependent bound as in Theorem 1.*

