# OpenReview forum: "Functional Critic Modeling for Provably Convergent Off-Policy Actor-Critic"
_ICLR.cc/2026/Conference — ICLR 2026 Conference Desk Rejected Submission_

### Official Review · Reviewer_szZh · 2025-10-28

**Soundness:** 3
**Presentation:** 2
**Contribution:** 2
**Rating:** 2
**Confidence:** 3

**Summary:**

The paper proposes a functional critic modeling approach for off-policy actor-critic reinforcement learning that aims to address the deadly triad instability and the challenge of estimating off-policy policy gradients. The authors provide theoretical convergence guarantees in the linear function approximation setting and demonstrate their method on continuous control tasks from the DeepMind Control Benchmark.

**Strengths:**

- Though not presented clearly (see my comments below), the idea of using functionals seems to fit well within the context of off-policy learning.
- The authors provide a few analyses that might be useful for others who want to apply functionals in other contexts.

- Some parts of the related work, specifically the earlier works in off-policy learning, are well-covered.

**Weaknesses:**

**Practical utility:** We've seen off-policy RL algorithms (e.g., TD3, SAC) excel in high-dimensional and continuous state-action spaces, even though they violate theoretical guarantees due to off-policy learning (e.g., the deadly triad) or function approximation of the critic (e.g., policy gradient compatibility conditions).

To achieve competitive performance in practice, the authors still use neural networks, which ultimately break the theoretical guarantees made under linear functionals anyway. Even in these settings, they don't present comparisons with state-of-the-art off-policy methods like TD3 and SAC (see also *Experiments* below).

The theory of off-policy learning and its guarantees have advanced considerably and are largely established. From this point forward, we should be able to see the practical advantage of any theoretical framework that claims to improve or guarantee off-policy learning.

Therefore, I'm skeptical about why one would need to use the authors' method when they can still use TD3 or SAC, given that theoretical guarantees already don't generalize to practice due to the use of neural networks. This is a critical gap I don't see addressed in the paper.

---

**Experiments:** The experiments are far from sufficient.

An immediate observation is the lack of proper statistical testing. The use of standard deviation isn't sufficient; the error bars are intertwined with each other in multiple environments, e.g., in Cheetah and Hopper (the 2 envs setting).

Why did the authors choose RLPD as the baseline method, which is meant to mix off-policy and offline learning, instead of direct off-policy algorithms? The main problem studied in the paper is the discrepancy introduced by off-policy learning. That said, I was expecting comparisons with TD3 and SAC, drawbacks of which primarily come from off-policy discrepancy. The inclusion of a (partially) offline learning method prevents the isolation of the off-policy benefits of the proposed method.

---

**Problem motivation:** In the introduction, the authors talk about "challenges" and "computational burdens," but don't explicitly mention what the real limitations of off-policy learning are. The problem formulation needs to be sharpened. See also the _Suggestions_ section below.

---

**Functionals:** I'm not familiar with the practical implementation of functionals (e.g., using large neural networks to take functions as arguments). From what I understand in the paper, one can realize a functional (say, the approximate critic) by passing the function's parameters (i.e., policy parameters), the functional's own parameters ($\xi$), and the functional's inputs (i.e., $s$ and $a$).

Is this something that has been tried in the literature? Has this been used before, even in other areas? No discussion or prior works on functionals are provided.

This is a significant gap in the paper that could make people unfamiliar with this approach (like me) question its practical effectiveness.

An immediate question is the study of the scale of functional parameters relative to the argument function's (i.e., the policy) parameters. I think this is a crucial aspect of the proposed method which isn't addressed in the paper.

---

**Implementation:** There are significant flaws in the presentation of the experiments.

First, to realize a functional, the authors use a large number of parameters (coming from the actor encoder initialized by a transformer). Applications of transformers in RL aren't explored/mentioned in the paper (even as a related work), yet the authors rely on transformer-based initialization without justification.

Second, the functional critics are initialized as "encoders," implemented by transformers. What is the input to the actor encoder? What kind of a transformer is used? These details are missing, but from what I understand from the reference to Devlin et al. is that the authors use bi-directional model. That said, the input to a bi-directional model should be a sequence, maybe the sequence of transitions, and the output embedding is supposed to be passed to the joint encoder. I don't think this is accurate in terms of implementation. The reason is that the actor encoder should take policy parameters per functional definition, but I'm unsure how this is possible in terms of transformers.

Third, additional yet substantial experimental details are missing. For instance, how many parameters are trained, and how would this compare to a regular actor-critic initialization with a standard hyperparameter setup? The use of functionals seems advantageous but requires significantly more parameters. There are no details about the used transformer encoder as well, which constitutes the backbone of the authors' idea. It's not clear if the authors' approach is still feasible even with the increased parameter count. A fair comparison would control for model capacity.

---

### Suggestions

- **Comparisons with TD3 and SAC:** I think we should see comparisons with the SOTA off-policy methods before we can deem the proposed method to be beneficial. It shouldn't take too long to implement and run the results for TD3 and SAC.

- **The last sentence of the first paragraph of Introduction:** Needs a reference; for example, to reinforcement learning from human feedback (where LLMs were first used as RL agents) or other seminal robotics work.

- **Second paragraph of Introduction, the last sentence:** How are the corrections "vulnerable to the same instability issues as the policy-evaluation step"? Providing a reference is not sufficient here. Please elaborate on this point.

- **Additional off-policy learning methods:** While no convergence rate was given, additional off-policy correction techniques that are gradient-free and use one-step TD learning were also proposed [1, 2]. The introduction might benefit from these additional citations.

- **Lines 288-289:** What are those "technical reasons"? This should be explained.

- **Typos:** Remove the closing curly bracket after the term $||\nabla_\phi(s, a; \theta)||$ in Assumption 2.

---

### Overall Assessment

While the idea of using functionals for off-policy learning is interesting, the paper suffers from several critical issues: (1) unclear practical utility given that existing methods like TD3/SAC already work well without theoretical guarantees, (2) insufficient experimental validation and lack of proper baselines, (3) missing implementation details and justification for design choices. The paper would benefit from a major revision addressing these concerns before it can be considered for acceptance.

---

**References**

[1] Cicek et al. _Off-policy correction for deep deterministic policy gradient algorithms via batch prioritized experience replay_. ICTAI 2021
[2] Saglam et al. _Mitigating Off-Policy Bias in Actor-Critic Methods with One-Step Q-learning: A Novel Correction Approach_. Transactions on Machine Learning Research, 2024.

**Questions:**

- **Line 426 – "Deterministic Actors":** The authors claim that stochastic exploratory policies come with several additional hyperparameters. Can the authors clarify what these parameters are? As far as I know, the exploratory policy is typically initialized as a Gaussian random variable with standard deviation 0.1 (or learned), and the mean is the deterministic action. I don't see sufficient justification for the claim there.

- **Third paragraph of Introduction:** "The two additional challenges" are mentioned but not particularly well explained. Specifically, what is meant by "estimation of several additional critic-like quantities"? Why are they difficult to estimate? Please clarify.

See also the weaknesses section.

---

### Official Review · Reviewer_6z2j · 2025-11-01

**Soundness:** 2
**Presentation:** 2
**Contribution:** 2
**Rating:** 2
**Confidence:** 4

**Summary:**

This work developed an actor-critic framework for off-policy reinforcement learning. The critic is functional in the sense that it takes the actor as input so that it can learn from different actors during training. Under certain assumptions, the paper developed convergence results for the algorithm. Through experiments on Cheetah-run and Hopper-hop continuous control tasks, it showed that the algorithm can work better than RLPD (Ball et al., 2023).

**Strengths:**

1. It developed an actor-critic framework with a functional critic so that it can generalize over different actors, which can be helpful for training if the critic has sufficient capacity

2. It analyzed the convergence property of the algorithm under certain assumptions in the linear function approximation setting

**Weaknesses:**

1. The main idea of having a functional critic that takes an actor as input has been explored before in different contexts, such as universal value function approximator (Schaul et al., 2015) or meta-critic (2017). These should be discussed and compared in the paper.

2. There are some problems in the theoretical analysis.

- It is not clear why the analysis from L309-329 is considered heuristic or informal.
- The C in Assumption 5 is unnecessary as the first term of Assumption 2 is enough.
- Assumption 6 requires further discussion. It is unclear why it would hold beyond simple settings.
- It is unclear why Theorem 2 is considered a convergence result. $b$ measures the bias of the gradient and the result shows that this bias is close to the gradient in size (norm), which does not make much sense as (1) it doesn’t really show convergence like vanishing gradient or (2) these two vectors can point to different directions yet have similar sizes. This result should definitely be discussed more and the paper should explain it’s interpretation and significance. Its proof should also be included for completeness.

3. Some algorithm designs need further explanation.

- The explanation of the actor encoders (L417-425) can be explained more clearly. $\zeta$ is said to be states (L419) but then mentioned to be parameters (L425). It would be clearer if a pictorial illustration is provided for the overall architecture.
- L445: Specifically, what is the “Thompson sampling flavor of design for exploration”? What theoretical properties does it have?

4. Experiments

- The experiment setting requires clarification. L130 said that the agent can only interact with a fixed behaviour policy $\mu$, as confirmed by line 3 of Algorithm 1. However, line 8 of Algorithm 3 allows the agent to interact with the environment with an arbitrary policy, which is "more on-policy".
- The algorithm is only compared to one baseline. There are several recent advancements (Ankile et al., 2025; Luo et al., 2024) that should at least be discussed if not compared.

Ref:

- Schaul, T., Horgan, D., Gregor, K. and Silver, D., 2015, June. Universal value function approximators. In *International conference on machine learning* (pp. 1312-1320). PMLR.
- Sung, F., Zhang, L., Xiang, T., Hospedales, T. and Yang, Y., 2017. Learning to learn: Meta-critic networks for sample efficient learning. *arXiv preprint arXiv:1706.09529*.
- Luo, Y., Ji, T., Sun, F., Zhang, J., Xu, H. and Zhan, X., 2024. Offline-boosted actor-critic: Adaptively blending optimal historical behaviors in deep off-policy rl. *arXiv preprint arXiv:2405.18520*.
- Ankile, L., Jiang, Z., Duan, R., Shi, G., Abbeel, P. and Nagabandi, A., 2025. Residual Off-Policy RL for Finetuning Behavior Cloning Policies. *arXiv preprint arXiv:2509.19301*.

Minor comments

- L126 $d_0$ but L127 $\mu_0$
- L133 & 134, throughout the paper twice
- L139 & 140, repeated sentences
- L215 $\pi_\theta$ should be $\theta$
- L275: There is a right bracket in the middle. Also, $\bar{\theta}$ should be $\theta’$
- L287: line 2 -> line 3
- L294: The truncation function is mathematically incorrect

**Questions:**

Q1: Why is L309-329 considered heuristic or informal?

Q2: Why is Theorem 2 considered a meaningful convergence result and how to interpret it?

Q3: Explain the architecture and Thompson sampling (see weakness #3 above).

Q4: Why does Algorithm 3 allow the agent to interact with the environment with an arbitrary policy?

---

### Official Review · Reviewer_CwwX · 2025-11-01

**Soundness:** 2
**Presentation:** 1
**Contribution:** 1
**Rating:** 2
**Confidence:** 3

**Summary:**

This paper proposes a new functional critic modeling framework for off-policy actor–critic reinforcement learning.
The central idea is to model the critic as a functional that maps a policy $\pi$ to its value function $Q_{\pi}(s,a)$, enabling the critic to generalize across changing policies without restarting evaluation.
The authors provide a convergence proof under a linear functional approximation assumption and claim this to be the first provably convergent off-policy target-based AC algorithm.
A minimal neural implementation with transformer-based actor encoders and deterministic actor ensembles is presented, showing preliminary results on DMC compared against RLPD.

**Strengths:**

1. The paper presents a clearly motivated study, addressing well-known challenges in off-policy actor–critic methods such as instability and the moving-target problem. It situates its contribution in relation to prior works like GTD, ETD, and target critic approaches.
2. The authors claim to be the first to offer a convergence guarantee for a target-based off-policy actor–critic algorithm, which, if validated, could be of theoretical value.

**Weaknesses:**

1. A substantial portion of the paper is dedicated to theoretical justification, but the experimental evaluation is limited. The empirical study compares only two tasks from DMControl against RLPD and lacks ablation studies, sensitivity analyses, or complexity/runtime comparisons. This makes it difficult to assess the practical relevance or efficacy of the proposed approach, especially whether performance gains are attributable to the functional critic itself. It would strengthen the work significantly to include more experiments probing convergence—e.g., through training with perturbed environments or replay buffers—to empirically demonstrate robustness to moving-target issues.
2. In terms of novelty, the paper’s contributions appear modest. Similar ideas have been explored under the names of meta-critic networks, hypernetworks, and parameter-conditioned critics. Moreover, as currently presented, Theorems 1 and 2 seem to be extensions of existing results, and the paper lacks a clear and convincing explanation of what is truly novel about its theoretical contribution.
3. The paper structure is incomplete. It would benefit from a more thorough discussion or conclusion section.

**Questions:**

1. Can you concretely demonstrate how the “functional critic” differs from a hypernetwork or meta-critic conditioned on actor parameters?
2. How sensitive is the proposed method to the choice of feature mapping ϕ(s,a,θ) in practice?
3. Can you provide ablations showing whether the transformer encoder is necessary?
4. Does the framework scale to high-dimensional or visual control tasks, or is it limited to DMControl setups?

---

### Official Review · Reviewer_HYnw · 2025-11-02

**Soundness:** 3
**Presentation:** 2
**Contribution:** 2
**Rating:** 2
**Confidence:** 3

**Summary:**

The paper proposes a “functional critic” that conditions the value function on the policy itself, i.e., learns $\hat Q(\pi_\theta, s, a; \xi)$ so that gradients $\nabla_\theta \hat Q$ provide an exact off-policy policy gradient without emphatic corrections or importance weighting. Concretely, they pair a policy (actor) encoder—built from “probing” the actor on a learned set of states—with a standard state–action encoder, fuse them in a joint critic, and train it via TD with a target-network scheme; the actor is updated by backpropagating through $\hat Q$ to $\theta$. On the theory side, under a linear functional approximation with a policy-dependent feature map and several regularity assumptions, they prove stability/convergence of a target-based actor–critic (Theorems 1–2). Empirically, on continuous-control benchmarks, the method looks competitive with strong off-policy AC baselines, with the claimed benefit that the critic generalizes across changing policies, reducing the mismatch that typically complicates off-policy actor updates.

**Strengths:**

- Clear formulation of a “policy-conditioned critic” and a concrete target-network AC instantiation.

- Potentially novel *use* of a policy-input critic to obtain an exact off-policy gradient via $\nabla_\theta \hat Q$ and a convergence argument for a target-based AC variant (within the stated linear setting).

- Practical section is detailed enough to reproduce the high-level recipe; empirical results are reasonable (if unsurprising).

**Weaknesses:**

- The manuscript asserts $ \sum_{s,a} d_{\pi_\theta}(s) \pi_\theta(a\mid s) \nabla_\theta Q_\theta(s,a)=0 $ (citing Sutton, 2000). I don't believe this is correct. In on-policy PG, differentiating the objective via chain rule would introduce three terms; the score-based term and two terms with $ \nabla_\theta d_{\pi_\theta} $ and $ \nabla_\theta Q_\theta $, since now the occupancy measure also depends on $\theta$. PGT avoids the chain rule route entirely and shows that gradient equals the score-based term, which implies that the two terms *sum to zero*, not that each term is zero.

- Prior research (Policy Evaluation Networks (PVN) [2] / network fingerprinting; Parameter-based Value Functions; PeVFA [1]; recent “scaling” papers [3] and a thesis on policy-conditioned value functions [4]) already input policy representations to value networks and study generalization. The submission does not cite or contrast with these, yet presents “functional critic modeling” as core novelty.

- Theory adds limited insight. Assumption 1 is essentially an oracle linearization that makes the results unsurprising; the remaining regularity assumptions rule out pathologies. The analysis does not cover the nonlinear case that matters most.

- “Practical Implementation” would benefit from a single figure wiring Eq. (9)/modules: actor encoder (probe states), state–action encoder, joint head, target copies, and gradient paths for $ \nabla_\theta \hat Q $.

- Ad-hoc design choices need explanation. Example: keeping the *same* actor encoder in the target path while delaying other encoders is not motivated.

- The way the functional critic is trained (policy-conditioned value network over a training set of policies) mirrors [1] and PVN/PVF.

- The actor representation is the same as the network fingerprinting approach in PVN. In PVN, policy network is fingerpinted by generating outputs from a set of "probing states" and this inputs-outputs vector is used to encode the policy network within the policy-conditioned value network. The probing states are also learned end-to-end. This is the same as the actor encoding used here.

- Since multiple actor-aware/policy-conditioned critic methods exist, at least one direct baseline from that family is missing.


[1] "What About Inputing Policy in Value Function: Policy Representation and Policy-extended Value Function Approximator" (AAAI 2022).

[2] "Policy Evaluation Networks" (2020, arXiv:2002.11833)

[3] "Massively Scaling Explicit Policy-conditioned Value Functions"(2025, arXiv:2502.11949)

[4] "Improving Policy-Conditioned Value Functions" (2022, https://github.com/Sebastian-Griesbach/Improving-Policy-Conditioned-Value-Functions/blob/main/thesis.pdf)

**Questions:**

1.  What is the precise novelty over PVN/PVF/PeVFA beyond the linear target-network convergence and the specific “exact off-policy gradient via $ \nabla_\theta \hat Q $” pathway? Please enumerate differences.
2. Why is the actor encoder *not* delayed in the target network while others are? What breaks if you delay it too?
3. How are probe states chosen/updated? How sensitive are results to the number of probe states and update frequency?
4. Do any parts of the analysis transfer to nonlinear critics (even as stability heuristics)? How does approximation error in the learned $ \phi $ propagate to bias in $ \nabla_\theta \hat Q $?
5. Could you add ablations: (i) remove policy input (standard critic), (ii) freeze actor encoder, (iii) shared vs. delayed actor encoder, (iv) deterministic vs. stochastic actors?
6. Given prior actor-aware critics, can you include at least one direct policy-conditioned-critic baseline for comparison?

---

### Note · Program_Chairs · 2026-01-17
**Submission Desk Rejected by Program Chairs**

The following references in this submission do not refer to real documents and/or have major errors in bibliographic information:

 Martha White. Developing a predictive theory of off-policy learning. arXiv preprint arXiv:1506.02476, 2015.